



# Significant soil degradation is associated with intensive vegetable cropping in subtropical area: A case study in southwest China

Ming Lu[1], David S. Powlson[2, 3], Yi Liang[1], Zhi Yao[1], Dave R. Chadwick[2, 4], Shengbi Long[5], Dunyi Liu[1]*, Xinping Chen[1, 2]

[1]College of Resources and Environment, Chongqing Key Laboratory of Efficient Utilization of Soil and Fertilizer Resources, Southwest University, Chongqing, 400715, China;
[2]Interdisciplinary Research Center for Agriculture Green Development in the Yangtze River Basin, Southwest University, Chongqing, 400715, China;
[3]Department of Sustainable Agriculture Systems, Rothamsted Research, Harpenden, Herts, AL5 2JQ, UK;
[4]School of Environment, Natural Resources and Geography, Bangor University, Gwynedd, LL57 2UW, UK;
[5]Jinping Station of Agricultural Technology Promotion, Guizhou, 556700, China.

*Correspondence to*: Dunyi Liu (liudy1989@swu.edu.cn)

**Abstract.** Within the context of sustainable development, soil degradation driven by land-use change is considered a serious global problem but conversion from growing cereals to vegetables is a change that has received little attention, especially in subtropical regions. Therefore, we compared the nutrient status and soil quality parameters (soil organic carbon [SOC], total nitrogen [TN], C/N ratio, pH, phosphorus [P], potassium [K], calcium [Ca], and magnesium [Mg]) between vegetable fields (VF) and land still used for paddy rice-oilseed rape rotation (PRF) that are typical of southwest China. In the VF, fertilizer application were often several times higher than the crop needs or recommended by the local extension service, thus, the crop use efficiency of N, P, K, Ca, Mg were only 26%, 8%, 56%, 23% and 28%, respectively; SOC, C stock, TN, N stock decreased significantly caused by low organic inputs from crop residues and high tillage frequency, and soil C/N ratio decreased slightly; available P (AP) in topsoil increased by 1.92 mg kg$^{-1}$ for every 100 kg ha$^{-1}$ of surplus P, and the critical levels of AP and CaCl$_2$-soluble P for P leaching were 104 and 0.80 mg P kg$^{-1}$. Besides, there was a clear trend of soil acidification in the VF. However, increasing concentrations of soil Ca and Mg significantly alleviated topsoil acidification, with the effect increasing over time. Given our findings, we discuss the potential benefits of conservation agricultural practices, integrated soil-crop system management strategies and agricultural technology services for recovering the degraded soil and improving the vegetable productivity in the VF.

## 1 Introduction

Land-use changes are an important anthropogenic perturbation that can cause soil degradation (Lal, 2015), but the impacts of land conversion from growing cereals to vegetables have received little attention. From 1997 to 2017, total vegetable production increased by 98% worldwide, and the area harvested increased by 59%. These increases were driven by economic growth and an increased demand for vegetables associated with a raised awareness of their health benefits (FAO, 2017; Yan



et al., 2012). The terrestrial expansion of vegetable production is particularly significant in the tropics and subtropics because these regions have a longer growing season compared to temperate regions (Fischer et al., 2012). In contrast to cereal crops, vegetable production requires more intensive management, including higher tillage frequencies and fertilization rates, and higher nutrient and water inputs due to the shallow root systems of most vegetable species (Wang et al, 2018a, b; Yan et al., 2013; Zhang et al., 2017). Thus, vegetable production is associated with a higher risk of environmental pollution. Zhang et al. (2012) found that the per unit nutrient inputs in vegetable production are 81-120% greater than those for cereal production system in China and 1.5-12 times greater than those for vegetable cropping in the rest of the world. In particular, the overuse or misuse of fertilizers in vegetable fields (VF) is a long-standing problem (Conley et al., 2009).

Soil organic carbon (SOC) is widely regarded to be the most reliable indicator for monitoring soil degradation (Lal, 2015). Bünemann et al. (2018) reviewed soil quality assessment approaches and summarized the indicators used; after SOC, the most frequently used chemical indicators were soil pH, available phosphorus (P), available potassium (K), total nitrogen (TN) and bulk density.

The impacts of land conversion from cereal to vegetable production may be more adverse in tropical and subtropical regions compared to temperate regions due to higher temperatures and rainfall combined with low organic inputs and high frequency of tillage practices (NAPCC, 2016; Powlson et al., 2016; Sarker et al., 2018). For instance, Wang et al. (2014) found that SOC concentration and the C/N ratio in open VF converted from paddy rice fields decreased by 19.7% and 27.8%, respectively, in southern China.

High inputs of N fertilizers to vegetable crops and the crop removal of base cations such as calcium (Ca) and magnesium (Mg) are expected to contribute to soil acidification (Guo et al., 2010; Zhang et al., 2016). However, in southwest China, fields receive large inputs of Ca and Mg from various fertilizers used for supplying micronutrients. To date, the combined impacts of these inputs and the removal of base cations on soil pH and other properties in vegetable systems have not been studied.

Systematic surveys to elucidate the effects of different cropping systems on soil properties are needed for a clear understanding of the impacts of vegetable production and the proper development of sustainable high-yield vegetable systems that do not cause environmental damage. In the present study, paddy rice-oilseed rape rotation fields (PRF) were used as a reference, because all VF in the region were converted from PRF. The objectives of this study were to (i) summarize the seasonal input-output apparent balance of N, P, K, Ca and Mg in PRF and VF, and (ii) assess the impacts of land-use conversion (from PRF to VF) on soil pH, SOC, TN, soil C/N ratio, available P (AP), $CaCl_2$-soluble P ($CaCl_2$-P), and available base cations (K, Ca, and Mg).



## 2 Materials and methods

### 2.1 Site descriptions and cropping systems

The field study sites were located in Jinping County in eastern Guizhou Province at 26° 23´ - 26° 46´ N and 108° 48´ - 109°
24´ E (Figure 1A). In this area, VF covered an area of 4,814 ha, representing an increase of 24% between 2009 and 2015,
and production increased by 132% between 2011 and 2018 (84,886 t). The area covered by cereal crops (14,408 ha at the
time of our study) decreased by 9% from 2009 to 2018. The region is characterized by a typical subtropical humid monsoon
climate with an annual average precipitation of 1160 mm for the 2007-2017 period, of which 85% occurs between April and
October, and a mean annual air temperature of 15.9°C (Figure 1B). The highest temperatures and rainfall occur in the same
season. The main soil type is yellow earth (Ultisol), and the main crops grown are open-field vegetables, paddy rice, oilseed
rape, and fruit trees.

Open-field vegetable production accounts for 70-80% of the total area of dryland agricultural production, and the most
common planting pattern (71% of the total VF area) is pepper-Chinese cabbage rotation system. In this cropping system,
peppers are transplanted in April and harvested in mid-August, then non-heading Chinese cabbage is grown continually for
three seasons, from August to January of the following year, followed by a fallow period until April (Figure 1B). During the
annual rain-fed cropping period, tillage is performed four times, and fertilizers are applied 10 times (four times for pepper,
and twice for each Chinese cabbage crop). Conventional farming practices commonly used throughout the region were used
for all crops. In the paddy rice–oilseed rape double-cropping system, paddy rice is grown from late May to September, and
then oilseed rape is grown from mid-October to early May of the following year.

### 2.2 Farmer surveys and soil sampling

In this study, the focus was only on the typical vegetable production (pepper -Chinese cabbage - Chinese cabbage - Chinese
cabbage - fallow rotation) and the paddy rice-oilseed rape rotation. Based on farmers' survey methodology (Jia et al., 2013),
to obtain representative results, two of the most important vegetable-production townships in Jinping County—Xinhua and
Dunzha—were selected for farmer surveys. In each township, four villages were randomly selected, and 15-20 farmers from
each village who had managed PRF before switching to VF were randomly surveyed. Survey questions were related to crop
varieties and the management of the two cropping systems, including sowing rate, fertilizer (chemical or organic) application
rate, tillage frequency, crop-residue management, crop growth cycle, cropping duration, and yield. A total of 24 surveys
were related to PRF management and 133 surveys to VF management.

To investigate the effects of long-term fertilization on nutrient surpluses, enrichment/depletion, and downward transport in
the soil profile in the VF, we collected soil samples randomly from 36 VF that were converted from PRF 1-3 years (n=12),
5-10 years (n=12), and ≥15 years (n=12) ago. Twelve adjacent PRF were also sampled for comparison; the soil type at all
sites was very similar. Soil samples were collected at six spots at 20-cm depth intervals to a depth of 60 cm within each plot
and pooled by depth. Samples were collected at all selected VF and PRF from September 25 to 31, 2018, after the crops



were harvested. In addition, three other soil cores were collected randomly from each field type at the same depths for bulk density measurements.

## 2.3 Soil analysis

In this study, the moist composite fields samples were air dried and passed through a 1-mm sieve for the analysis of soil pH; $CaCl_2$-P; AP (determined using the Bray method); water-soluble K, Ca, and Mg; and exchangeable K, Ca, and Mg. The samples were then passed through a 0.15-mm sieve for SOC and TN analysis. Soil pH in a 1:2.5 soil-to-water suspension was measured using a potentiometer, SOC was measured using dichromate digestion, and TN was measured using the semi-automatic Kjeldahl method. $CaCl_2$-P was extracted from a 1:5 soil-to-water suspension using 0.01 mol L$^{-1}$ $CaCl_2$, AP was extracted using 0.03 mol L$^{-1}$ NH$_4$F and 0.025 mol L$^{-1}$ HCl, water-soluble K, Ca, and Mg were extracted in the water portion of a 1:5 soil-to-water suspension, and exchangeable K, Ca, and Mg were extracted with 1 mol L$^{-1}$ neutral NH$_4$OAC (Lu, 2000).

## 2.4 Calculation of nutrient inputs and balance

Detailed records were made of the farm management practices employed in each field from 2017 to 2018, and we confirmed that the management history of each field was similar to the management practices employed during the survey year. The main mineral fertilizers used in this region were urea, potassium sulfate, compound or mixed fertilizers, calcium super-phosphate and fused calcium-magnesium phosphate. Organic fertilizers included crop residues and commercial organic fertilizer, animal manure (from chickens, cows, and pigs), and rapeseed meal. In the PRF, rice straw was directly returned after the grains were harvested, whereas oilseed rape straw was burned in the fields. Most above- and below-ground residues in the VF were removed during harvest.

Nutrient removal due to grain harvest or the removal of aboveground crop parts was calculated using yields reported by farmers and published nutrient concentrations typical of the region. Nutrient surpluses were calculated by subtracting nutrient removal due to grain harvest or aboveground removal from total fertilizer inputs.

## 2.5 Calculation of SOC and TN stock, accumulation of P surplus and changes in soil AP

The SOC and TN stocks in the PRF and VF soils were calculated by multiplying SOC and TN concentrations with bulk density and depth. Accumulation of surplus P was calculated as the annual P surplus multiplied by planting duration, and changes in soil AP in the VF were calculated as soil AP in the PRF subtracted from that in the VF.

## 2.6 Statistical analysis

SPSS software (ver. 21.0 for Windows; IBM Corporation, Armonk, NY, USA) was used for statistical analysis. Differences in soil nutrient concentrations and stock among cropping systems were assessed using one-way analysis of variance, and mean values were compared using the least significant difference test with a significance level of 5%. Correlations were



analyzed using a two-tailed Pearson test with a significance level of 5%. The linear-plateau model in SAS (ver. 9.2 for Windows; SAS Institute, Cary, NC, USA) was used to analyse the relationship between SOC and tillage frequency in the 0–20-cm soil layer in the VF, and the two-segment linear model in SigmaPlot (ver. 12.5 for Windows; Systat Software, San Jose, CA, USA) was used to analyse the relationship between soil AP and $CaCl_2$-P.

## 3 Results

### 3.1 Yields, fertilizer use and nutrient balance in two intensive cropping systems

Total annual inputs of N, P, Ca, and Mg in the VF were several times higher than those in the PRF (2.38, 2.97, 4.40 and 7.14 times, respectively), but the total annual K inputs were similar (Table 1). K, Ca, and Mg inputs in the PRF were mainly from organic sources, i.e., a combination of organic fertilizers and crop residues, representing 65%, 80% and 65% of the total inputs, respectively, while chemical fertilizer supplied 61% and 72% of total N and P inputs. In contrast, in the VF, these nutrients were mainly supplied by chemical fertilizers, which comprised 75%, 86%, 71%, 75%, and 79% of the total inputs for N, P, K, Ca, and Mg, respectively. Fertilizer application rates did not differ significantly among VF according to the time since conversion from PRF (Table 1). Vegetable yields also did not differ significantly.

In the PRF, straw was returned to the field after grain harvest, whereas in the VF, all plant biomass was completely removed during harvest. Consequently, 2.46, 9.05, 14.0, and 4.52 times more N, K, Ca, and Mg were removed, respectively, from VF than from PRF. However, the P removal rate in the VF was slightly lower. Large surpluses of N (790 kg N ha$^{-1}$ yr$^{-1}$), P (343 kg P ha$^{-1}$ yr$^{-1}$), and K (235 kg K ha$^{-1}$ yr$^{-1}$) were found in the VF. Unexpectedly, there were also large surpluses of Ca (800 kg Ca ha$^{-1}$ yr$^{-1}$) and Mg (161 kg Mg ha$^{-1}$ yr$^{-1}$) in the VF compared to the PRF (Table 1).

### 3.2 Changes in SOC and TN in VF

The concentration of SOC, TN, C/N ratio, C stock and N stock were compared between VF and PRF (Table 2). Conversion from PRF to VF led to a significant decrease in both TN concentration and stock in the topsoil (0-20 cm), with mean decreases of 15.8% and 17.6% ($P < 0.05$), respectively. However, conversion to VF did not result in significant changes in TN in the two deeper soil layers. Conversion from PRF to VF led to significant decrease in topsoil SOC concentration and stock, even after only 1-3 years. The C stock in the 0-20 cm soil layer decreased by 10.3% after 1-3 years and increased to 24.3% after 5-10 years. Similar trends were observed in the 20-40 cm and 40-60 cm soil layers. Because proportional decreases in SOC were slightly greater than those for TN, the soil C/N ratio became slightly smaller following conversion to VF, with this trend most apparent in the 40-60 cm soil layer and decreased by 24.3% after 5-10 years.

The average bulk density in the topsoil tended to decrease upon conversion from PRF to VF but did not change significantly according to years of vegetable production. Averaged over all periods of VF, C stock in the 0-60 cm soil layer was 15.9% ($P < 0.05$) less in the VF than in the PRF with values of 94.3 versus 112 Mg C ha$^{-1}$ yr$^{-1}$.



One major reason SOC might have decreased in the VF was that the organic C input in the PRF was 1.35-fold higher than that in the VF (Figure 2A). Tillage practices were also performed more frequently in the VF, which supported four crops per year instead of two in the PRF. Tillage frequency, which was as high as 4 times per year, clearly affected topsoil SOC (0-20 cm) but not that in deeper soil layers (Figure 2B). Moreover, despite the large N surplus (790 kg N ha$^{-1}$, assumed to be predominantly nitrate) in VF, soil N stock generally decreased in line with SOC stock, with a slight variation to this trend observed in the 40-60cm layer (Figure 2C).

### 3.3 Available P changes in VF

Soil AP and CaCl$_2$-P in the 0-60 cm soil layer were significantly higher in the VF than in the PRF, with levels increasing according to time since VF conversion (Figure 3). For example, in VF converted >15 years ago, soil AP concentration was 22-fold those in PRF and CaCl$_2$-P in the topsoil was 9-fold higher. The increases in deeper soil layers are indicative of downward movement of P. In the 0-20 cm soil layer in the VF, the amount of surplus P accumulated (P inputs minus crop offtakes) was significantly and positively correlated with differences in soil AP between VF and PRF at the end of the growing season (VF, pepper season; PRF, paddy rice season) (Figure 4A). For every 100 kg ha$^{-1}$ of surplus P in the VF, AP concentration in the 0-20 cm soil layer increased by 1.92 mg kg$^{-1}$. The phenomenon of a "change-point" has previously been observed in the relationship between soil AP and CaCl$_2$-P (Hesketh and Brookes, 2000), and used as an indicator of the risk of P loss. In VF topsoil(0-20 cm layer), AP and CaCl$_2$-P values at the change point as calculated using a two-segment linear model were 104 and 0.80 mg P kg$^{-1}$, respectively (Figure 4B). Above the change-point, soil CaCl$_2$-P increased rapidly with increasing AP.

### 3.4 Soil available K, Ca, and Mg in VF

The K nutrient removed via harvested crops in the PRF (33.8 kg K ha$^{-1}$ yr$^{-1}$) was only 11% of that removed from the VF (average, 306 kg K ha$^{-1}$ yr$^{-1}$) (Table 1). Thus, the K surplus in the PRF was, on average, 2.36 times that in the VF. However, soil available K levels (water-soluble and exchangeable fractions) were considerably higher in the VF in all three soil layers (Figure 5A and 5D). The depth distribution of soil available K indicated that there was substantial downward movement of K. The accumulation of available K, but not exchangeable K, in each soil layer increased with the number of planting years since VF conversion. Water-soluble K concentrations in the 0-20, 20-40, 40-60 cm soil layers in VF that had been continuously cultivated for over 15 years were 26%, 40%, and 275% higher, respectively, than those in VF cultivated for 1-3 years, and 193%, 180% and 650% higher, respectively, than those in PRF.

Water-soluble Ca was significantly higher in the VF (0-60 cm) than in the PRF soil (Figure 5B). Levels increased in the first 10 years after conversion from PRF but not thereafter. Exchangeable Ca in the topsoil (0-20 cm) also increased with time since VF conversion (Figure 5E). Soil available Mg (water-soluble and exchangeable fractions) also increased after conversion from PRF to VF and followed the same general trends as for Ca (Figure 5C and 5F).





## 3.5 Changes in soil pH in VF and effect factors

In the PRF, the soil was slightly acidic, especially in the topsoil (0-20 cm), with an average pH of 5.34 across the fields sampled, and initially decreased to approximately pH 4.7 after conversion to VF but recovered eventually (Figure 6). The initial pH was higher in the deeper soils (~pH 6.5) than in the 0-20 cm layer in the PRF, but it was significantly lower in the VF. The concentration of soil exchangeable Ca and soil pH were significantly and positively correlated in both the VF and PRF, particularly in the VF (Table 3). Soil exchangeable Mg exhibited the same trend in the VF but not in the PRF. In
addition, excessive N and K surpluses were significantly correlated with decreased soil pH in the 0-20 cm soil layer. Conversely, the enrichment of soil P had a negative effect on soil pH.

## 4 Discussion

### 4.1 Nutrient balance

Through the farmer surveys, we obtained a considerable amount of data on the fertilizer application rates employed in PRF
and VF, the two main cropping systems used in the southwest China. The annual application rates of fertilizer N, P and K in the VF (Table 1) were much higher than those recommended by the extension service (N, 600-765, P, 79-144, K, 398- 498 kg ha$^{-1}$) (Zhang et al., 2009). In the PRF, fertilizer N and K inputs were also much higher than the recommended (N 255- 330, K 112- 174 kg ha$^{-1}$), but the annual fertilizer P rate was close to the local recommended rate (P 33- 79 kg ha$^{-1}$) (Zhang et al., 2009). Consequently, nutrient surpluses developed under both cropping systems but were especially large in VF. Thus,
nutrient use efficiency (nutrients removed in the harvested crop expressed as a percentage of nutrient input) in the VF was very low. On average, the N, P, K, Ca, and Mg use efficiencies were 26%, 8%, 56%, 23%, and 28%, respectively. Studies assessing vegetable crops at the national scale have reported that the N use efficiency in China is 26 ± 13%, consistent with our finding (Ti et al., 2015). The low P use efficiency found for vegetable crops in the present study is even lower than the values reported in previous studies (15-20%) (Conley et al., 2009; Yan et al., 2013; Zhang et al., 2008).
Crop yields and nutrient use efficiency in VF in southwest China can be limited by a lack of support from local extension services (Ju et al., 2007), including a lack of fertilization recommendations or publicity materials, remoteness of advisers from the production fields, and a lack of training for most farmers. The average farm size in this region (calculated as the weighted mean) is 0.07 to 0.1 ha, and limitations can also be due to small farm size (Zhang, 2017). Farm size is often used as a proxy for the level of professionalism exhibited by farmers and awareness of efficient nutrient-management practices (Ju et
al., 2016; Wu et al., 2018). Therefore, smallholder farmers in China generally have a low awareness of fertilizer-saving practices, and farm size significantly influences the usage intensity and efficiency of agricultural chemicals.

Previous work by our group has shown that there is considerable scope for increasing yields and decreasing fertilizer use in vegetable production in this region (Wang, 2018b). Integrated soil-crop system management approaches (ISSM) based on advanced crop and nutrient management have been successfully developed for cereal cropping in China (Chen et al., 2014).



The grain yields of maize, rice and wheat can be increased by 35% (n=5,406), 18% (n=6,592) and 24% (n=6,940),
respectively, with no increase in fertilizer inputs and often, a decrease instead, using an ISSM approach (Chen et al., 2014).
The integrated management system has been developed for greenhouse pepper production in the Yangtze River Basin. After
2 years, pepper yield increased by 16% and N application rates decreased by 54%, due to the optimization of soil
management, plant population, and nutrient management (Wang et al., 2020). Thus, local extension services could provide

support and resources for applying integrated soil-crop system management methods to increase vegetable production,
minimise nutrients surpluses, and mitigate soil degradation in VF in the region.

**4.2 SOC and N stocks**

This study provides unequivocal evidence that converting land from long-established PRF to VF sharply decreased SOC and
TN concentrations and stocks. The trend was clearest in the topsoil where initial levels of SOC and TN were the highest, but

it was also apparent in the 20-40 and 40-60 cm soil layers (Table 2). SOC tended to be depleted further with an increasing
period of vegetable production, particularly in the first 10 years (Table 2), but decreases were generally small thereafter. Low
input of organic C from crop residues (Figure 2A) and a high tillage frequency (Figure 2B) were two responsible factors
caused that. The results agreed with the findings of Wang et al. (2014), who reported that SOC concentration decreased by
25% following conversion from paddy rice to vegetable production in southern China. We estimated that the average annual

C input in VF was less than 75% of that in PRF (Figure 2A). A higher frequency of tillage practice results in the break-up of
soil macro-aggregates, soil structure damage (Chivenge et al., 2007; Pires et al., 2016; Six et al., 2000), and an increase in
soil aeration (Sarker et al., 2018), which promotes microbial decomposition of soil organic matter. Decomposition is further
accelerated by high air temperatures and soil moisture levels in sub-tropical regions compared to temperate regions (Zhang
et al., 2013).

Under conventional VF management, at least four tillage operations per year are required. In contrast, an alternative strategy
is conservation agriculture (CA), which can create a positive soil/ecosystem C budget that often increases agricultural
productivity while sequestering additional C into the C stock, or at least, minimises SOC depletion (Chivenge et al., 2007;
Corbeels et al., 2019; Lal et al., 2004; Soussana et al., 2019). CA is premised on four basic principles (Lal, 2015): (i)
retention of crop residues as mulch, (ii) incorporation of a cover crop in the rotation cycle, (iii) use of ISSM, including

chemical and organic fertilizers, and (iv) minimal mechanical and manual soil disturbances. The performance of CA
strategies has been validated in the US Corn Belt (Clay et al., 2012) and sub-Saharan Africa (Vanlauwe, 2012) in terms of
increasing or maintaining SOC content. Declines in SOC and TN in VF in southwest China can also be slowed by including
a cereal crop in the rotation and incorporating straw into the soil. However, it should be noted that the high temperatures and
moisture levels prevalent under sub-tropical conditions lead to the fast decomposition of added organic inputs. For example,

in southwest China, the sequestration efficiencies of C from poultry manure, ruminant manure, compost, green manure, and
crop straw were only approximately 15% (Huang et al., 2018; Zhao et al., 2016).



Our results also indicated that soil TN content declined proportionally less than SOC content after conversion to VF. This was accompanied by marked changes in the soil C/N ratio (Table 2). For instance, in the 20-40 cm layer depth, the C/N ratio decreased from 10.5 in PRF to 9.01 in VF where vegetable production had been practiced for >15 years, and a larger change was observed in the 40-60 cm soil layer. Although the changes were significant, the absolute quantities of C and N in the deeper soil layers were small (around 7-8 g C kg$^{-1}$ soil and 0.9 g N kg$^{-1}$ soil; Table 2); thus, experimental error from sampling and analysis could be easily magnified when calculating ratios. However, the C/N ratio did seem to exhibit a decreasing trend, which has been noted in other studies of land-use conversion from cereal to vegetable cropping (Ju et al., 2007; Yan et al., 2012). Because of the very large N surplus in the VF (mean over three vegetable cropping periods, 789 kg N ha$^{-1}$ yr$^{-1}$, Table 1), it is perhaps conceivable that a fraction of the inorganic N in the soil profile was immobilized into soil organic matter under the conditions prevailing in the VF, thus counteracting the loss of N that accompanied the observed SOC loss. The slightly lower pH in subsoil in the VF compared to the PRF might have slowed nitrification such that part of the inorganic N remained in the ammonium form sufficiently long enough for some immobilization to occur (Figure 6).

We did not directly measure N losses at the study sites, but the large N surpluses strongly imply that the losses would have been substantial. Zhang et al (2017) had found that 36 % of the N applied (at a rate of 139 kg N ha$^{-1}$) during the growing season of bitter gourd was lost by leaching. In another intensive vegetable-cropping system, Wang et al (2018a) and Wang et al (2019) measured a loss of 3.91 kg $N_2O$-N ha$^{-1}$ season$^{-1}$ and a runoff of 16.5 kg N ha$^{-1}$ yr$^{-1}$. Therefore, losses of this order are to be expected in our study region. Hence, to minimize N loss, which leads to undesirable environmental impacts such as water pollution, eutrophication, and greenhouse gas emissions, rational N fertilizer management for intensive vegetable production in high-rainfall, sub-tropical environments is urgently needed.

**4.3 P enrichment**

P enrichment in farmland soils in China has been a serious problem for many years, as the movement of phosphate to surface waters causes environmental pollution (Bai et al., 2013; Cao et al., 2012; Li et al., 2011; Yan et al., 2013). On average, the annual P input (373 kg P ha$^{-1}$; Table 1) in the present study was 2.6~4.7 -fold higher than those recommended for VF (Zhang et al., 2009). Soil AP varied widely and ranged from 14.9 to 229 mg P kg$^{-1}$ (mean, 120 mg P kg$^{-1}$) in the 0-20 cm soil layer with a large variation, which is several times higher than the critical AP based on yield for fruit vegetables (58 mg P kg$^{-1}$) and leafy vegetables (46 mg P kg$^{-1}$) (Yan et al., 2013). The AP concentration in the topsoil (0-20 cm) increased by 1.92 mg kg$^{-1}$ for every 100 kg ha$^{-1}$ of surplus P in the VF. In other soils and cropping systems, the rate of increase ranges from 1.44 to 5.74 mg kg$^{-1}$ (Cao et al., 2012). A high level of P enrichment in the soil is detrimental to plant growth because it inhibits the rhizosphere manipulation processes employed by plants to efficiently acquire P, including the colonization of roots by arbuscular mycorrhiza fungi and the exudation of organic acids or phosphatase enzymes (Deng et al, 2017; Dong et al., 2004; Ryan and Graham, 2002). In addition, $CaCl_2$-P serves as an indicator of the risk of P loss from leaching (Hesketh and Brookes, 2000). Previous studies have reported a "change-point" in the graph of soil available P against $CaCl_2$-P, above which the risk of P leaching increases greatly (Bai et al., 2013; Shen et al., 2018; Yan et al., 2013). Here, this value was



exceeded in the 0-20 cm soil layer in many VF. As such, excessive P fertilizer application should be avoided to reduce the P surplus to zero and P movement to water bodies.

**4.4 Soil pH respond to land use change**

At the national scale, long-term intensive farming has been shown to cause soil acidification, with an average decline of 0.50 in pH in the topsoil of major Chinese croplands from 1980 to 2000 (Guo et al., 2010). Soil acidification is caused by the
excessive application of N fertilizers and the loss of base cations via leaching (Guo et al., 2010; Zhang et al., 2016), and leads to decreased crop production and quality (Cakmak and Yazici, 2010). After the conversion from PRF to VF, soil pH at all depths decreased, especially at 1-3 years after conversion (Figure 6). After this period, soil pH at all depths decreased, especially at 1-3 years after conversion. After this period, the pH in the 0-20 cm soil layer increased slightly and either stayed constant or decreased very slightly in the deeper soil layers. In the VF, exchangeable Mg and, to a lesser extent, Ca,
accumulated due to the types of fertilizers used. This apparently counteracted the trend towards acidification, as shown by the correlation between Mg and Ca accumulation and soil pH (Table 3). In this sense, the current fertilization practice does confer benefits in terms of improving soil quality. However, the greater accumulation rate of Mg compared to Ca is likely to cause physiological damage to crops due to the antagonism between the two elements. An imbalance in mineral nutrients in VF soil has been observed in other fields in southwest China.

The mean soil exchangeable Mg concentration in VF converted 1-3 years (73.2 mg kg$^{-1}$) is considered deficient, as it is much lower than the critical threshold (Bai et al., 2004). Cationic antagonism is also an important influencing factor that influences mineral nutrient bioavailability (Tisdale et al., 1993; Marschner, 2012). With the increase of planting duration (1-10 yrs), soil exchangeable K, Ca and Mg concentration gradually increased (Figure 5); but the growth ratio of each nutrient was not consistent (K > Mg > Ca), and the ratio of Ex-K/Ex-Mg decreased from 1.92 to 1.37 can be observed, while the ratio Ex-
Ca/Ex-Mg of remained at 6.14-6.29, with little change. However, it has been reported that plant Mg absorption would be inhibited by soil K and Ca when the value of soil Ex-K/Ex-Mg or Ex-Ca/Ex-Mg greater than 0.6 or 7.0, respectively (Tisdale et al., 1993; Morton et al., 2008). Thus, current vegetable-cropping practices can easily result in Mg physiological deficiencies (Yan et al., 2016). Nowadays, more and more attention has been paid to Mg issues in soil-plant-human continuum, and this intensive vegetable rotation in southwest China urgently needs optimal Mg fertilization strategy.

**5 Conclusions**

Open-field vegetable-cropping practices in southwest China can result in significant soil degradation and potential environmental pollution after conversion from a paddy rice-oilseed rape rotation system. Fertilizer application rates in VF are often several times higher than those needed to meet crop requirements or recommended by the local extension service.
Excessive fertilization has led to the substantial soil enrichment of P, K, Ca, and Mg. We also observed a clear trend of soil

acidification at 0-60 cm soil layer depths, although this was partially alleviated by the accumulation of Mg and Ca. SOC decreased rapidly after conversion to VF, and levels continued to decrease over time. This was attributed to decreased inputs from crop residues compared to PRF and a high tillage frequency, as well as the high temperatures and moisture levels prevalent under subtropical climatic conditions. The loss of SOC inevitably causes a decline in soil physical and biological

properties, with potential long-term impacts on the sustainability of the vegetable-cropping system. The C/N ratio tended to decrease in deeper soils layers in VF. Management practices likely to slow or reverse soil degradation and prevent and environmental pollution in the vegetable-cropping system include increasing organic inputs (from manure, organic fertilizers, or crop residues; or including cereals in the crop rotation so as to incorporate straw), decreasing fertilizer application rates to better match crop requirements, decreasing tillage frequency, and changing the types of chemical fertilizers used to avoid an

imbalance between Mg and Ca. These may be achieved by applying ISSM concepts and CA practices as well as through the development of effective strategies to deliver information to farmers.

**Data availability**

The data that support the findings of this study are available by request from the corresponding author (D. Liu).

**Author contributions**

DL and XC designed the experimental setup. ML, YL and YZ did soil sampling, led the lab analysis procedure and the farmer survey with the input of SL. ML, DP and DL also did the statistics, prepared the manuscript with valuable contributions of DC and XC, and undertook the revisions during the review process.

**Competing interests**

The authors declare that they have no conflict of interest.

**Financial support**

This research is supported by funding from the National Key Research and Development Program of China (No. 2018YFD0800600), the National Natural Science Foundation of China (No.31902117), the China Postdoctoral Science Foundation (No. 2018M643393), the State Cultivation Base of Eco-agriculture for Southwest Mountainous Land (Southwest University), and UK Research and Innovation (UKRI) through the UK Biotechnology and Biological Sciences Research

Council (BBSRC).



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




**Table 1: Average annual nutrient inputs, outputs, and surpluses in paddy rice–oilseed rape rotation fields (PRF) and vegetable fields (VF) converted from PRF 1-3, 5-10, or ≥ 15 years ago in southwest China.**

| Cropping system | PRF | VF | | |
|---|---|---|---|---|
| | | 1-3 years | 5-10 years | ≥15 years |
| **Inputs** (kg ha$^{-1}$ year$^{-1}$) | | | | |
| Organic fertilizers [a] | | | | |
| N | 175 | 314 | 151 | 344 |
| P | 35.2 | 69.7 | 32.5 | 58.8 |
| K | 383 | 155 | 101 | 210 |
| Ca | 188 | 262 | 144 | 388 |
| Mg | 20.4 | 50.7 | 25.1 | 65.4 |
| Chemical fertilizers | | | | |
| N | 272 | 843 | 882 | 672 |
| P | 90.6 | 305 | 399 | 256 |
| K | 206 | 480 | 363 | 315 |
| Ca | 47.9 | 457 | 1271 | 598 |
| Mg | 10.8 | 103 | 287 | 135 |
| Total fertilizers | | | | |
| N | 447 | 1153 | 1030 | 1013 |
| P | 125 | 375 | 431 | 314 |
| K | 589 | 634 | 463 | 526 |
| Ca | 236 | 719 | 1414 | 985 |
| Mg | 31.1 | 153 | 311 | 200 |
| **Outputs** (year$^{-1}$) | | | | |
| Crop yield (Mg ha$^{-1}$) [b] | 8.69 | 99.5 | 81.3 | 95.6 |
| Nutrient removal at harvest (kg ha$^{-1}$) [c] | | | | |
| N | 112 | 287 | 255 | 284 |
| P | 41.7 | 32.0 | 27.5 | 31.4 |
| K | 33.8 | 321 | 281 | 316 |
| Ca | 17.1 | 261 | 208 | 248 |
| Mg | 13.6 | 65.1 | 55.8 | 63.5 |
| **Nutrient surplus** (kg ha$^{-1}$ year$^{-1}$) | | | | |
| N | 335 | 866 | 775 | 728 |
| P | 84.0 | 343 | 404 | 283 |
| K | 555 | 313 | 182 | 210 |
| Ca | 219 | 458 | 1206 | 737 |
| Mg | 17.6 | 88.6 | 256 | 137 |

[a] Organic fertilizers included crop residues, commercial organic fertilizers, animal manure (from chickens, cows, and pigs), and rapeseed meal. In PRF, rice straw was directly returned after the grains were harvested, whereas oilseed rape straw was burned in the fields. Most above- and below-ground residues in the VF were removed during harvest.

[b] Vegetable yields are expressed as the annual sum of fresh weights per season, and yields in the PRF are expressed as the annual sum of air-dried weights per season.

[c] Only grains were harvested in the PRF, and most of the above- and below-ground residues were returned to the fields. All vegetable parts were removed from the fields during harvest.





**Table 2: Soil organic carbon (SOC) and total nitrogen (TN) concentrations, soil C/N ratio, and C and N stocks in the 0-20, 20-40,**
**and 40-60 cm soil layers in paddy rice-oilseed rape rotation fields (PRF) and vegetable fields (VF) converted from PRF 1-3, 5-10,**
**or ≥ 15 years ago in southwest China.**

| Soil depth (cm) | Cropping system | SOC (g C kg$^{-1}$) | C stock (Mg C ha$^{-1}$) | TN (g N kg$^{-1}$) | N stock (Mg N ha$^{-1}$) | Soil C/N ratio |
|---|---|---|---|---|---|---|
| 0-20 | PRF | 21.5 a | 55.6 a | 1.95 a | 5.04 a | 11.1 a |
| | 1-3$_{yrs}$VF | 19.6 b | 49.9 b | 1.78 b | 4.53 b | 11.1 a |
| | 5-10$_{yrs}$VF | 16.7 c | 42.1 c | 1.54 c | 3.89 c | 10.9 b |
| | ≥15$_{yrs}$VF | 16.9 c | 42.7 c | 1.59 c | 4.03 c | 10.6 b |
| 20-40 | PRF | 12.8 a | 33.8 a | 1.22 a | 3.16 a | 10.5 a |
| | 1-3$_{yrs}$VF | 12.3 ab | 32.1 ab | 1.25 a | 3.18 a | 9.83 b |
| | 5-10$_{yrs}$VF | 11.6 ab | 30.1 b | 1.18 a | 2.98 a | 9.77 b |
| | ≥15$_{yrs}$VF | 11.3 b | 29.2 b | 1.25 a | 3.16 a | 9.01 c |
| 40-60 | PRF | 8.67 a | 22.8 a | 0.89 b | 2.31 a | 9.84 a |
| | 1-3$_{yrs}$VF | 6.98 b | 18.2 b | 0.94 a | 2.40 a | 7.41 b |
| | 5-10$_{yrs}$VF | 7.77 ab | 20.2 b | 0.93 ab | 2.34 a | 8.44 b |
| | ≥15$_{yrs}$VF | 7.02 b | 18.3 b | 0.98 a | 2.47 a | 7.27 b |

Values were averaged over twelve replicates. The least significant difference test was performed to compared values in each soil layer, and

different lowercase letters indicate significant differences at $P < 0.05$ among cropping systems.



**Table 3: Pearson correlation analysis of the relationships between soil properties and soil pH in the 0-20, 20-40, and 40-60 cm soil layers in vegetable fields (VF) and paddy rice-oilseed rape rotation fields (PRF).**

| Soil depth (cm) | SOC | TN | C/N ratio | AP | Ws-K | Ws-Ca | Ws-Mg | Ex-K | Ex-Ca | Ex-Mg | N surplus | P surplus | K surplus | Ca surplus | Mg surplus |
|---|---|---|---|---|---|---|---|---|---|---|---|---|---|---|---|
| VF (n = 36) | | | | | | | | | | | | | | | |
| 0-20 cm | -0.46** | -0.45** | 0.10 | 0.30 | -0.32 | 0.22 | 0.27 | -0.01 | 0.89** | 0.81** | -0.48** | -0.09 | -0.45** | 0.32 | 0.28 |
| 20-40 cm | -0.40 | -0.54** | 0.05 | -0.37* | -0.63** | -0.13 | -0.04 | -0.37* | 0.58** | 0.58** | 0.10 | 0.14 | 0.03 | 0.05 | 0.06 |
| 40-60 cm | -0.313 | -0.02 | -0.34* | -0.44* | -0.25 | -0.39* | -0.39* | -0.38* | 0.480** | 0.40* | 0.39* | 0.23 | 0.27 | -0.11 | -0.08 |
| PRF (n =12) | | | | | | | | | | | | | | | |
| 0-20 cm | 0.21 | 0.35 | -0.22 | -0.13 | 0.47 | 0.57 | 0.08 | -0.05 | 0.59* | 0.29 | - | - | - | - | - |
| 20-40 cm | 0.30 | 0.51 | 0.69* | -0.07 | 0.32 | 0.81** | 0.15 | 0.35 | 0.66* | -0.10 | - | - | - | - | - |
| 40-60 cm | 0.05 | 0.01 | 0.02 | -0.48 | 0.17 | 0.65* | 0.06 | 0.19 | 0.72** | -0.43 | - | - | - | - | - |

SOC, soil organic carbon; TN, total nitrogen; AP, soil available phosphorus; Ws-K (Ca, Mg), soil water-soluble potassium (calcium, magnesium); Ex-K (Ca, Mg), soil exchangeable K (Ca, Mg; includes the water-soluble fraction). Significance levels *, $P < 0.05$; **, $P < $

0.01.



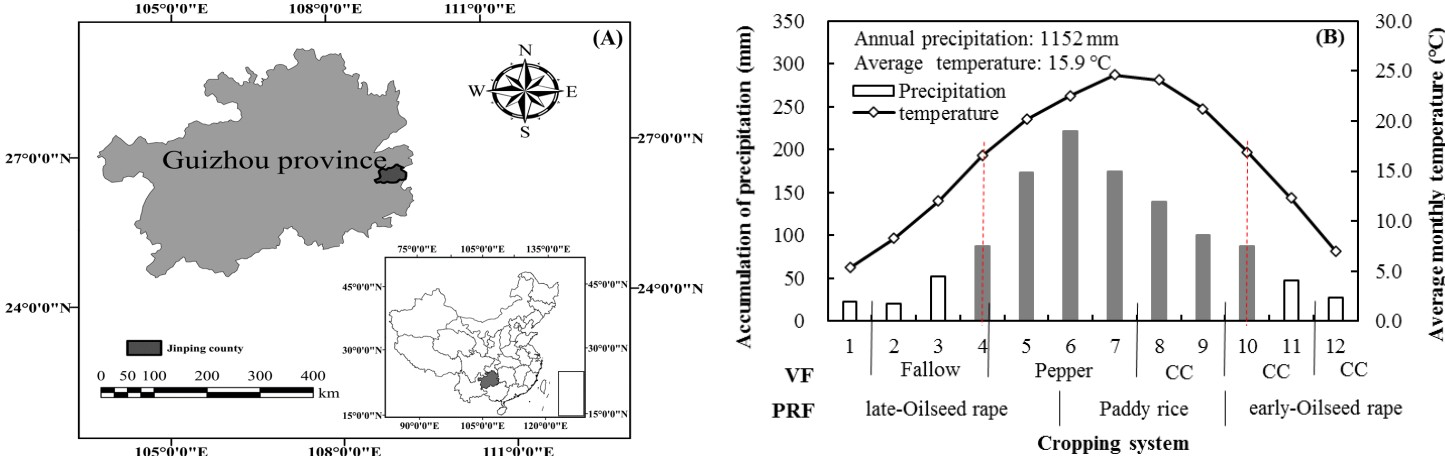

**Figure 1: The Location (A), meteorological conditions (values averaged between 2007 and 2017), and typical cropping systems (B) of the study area. CC, non-heading Chinese cabbage; VF, vegetable fields; PRF, paddy rice-oilseed rape rotation fields.**





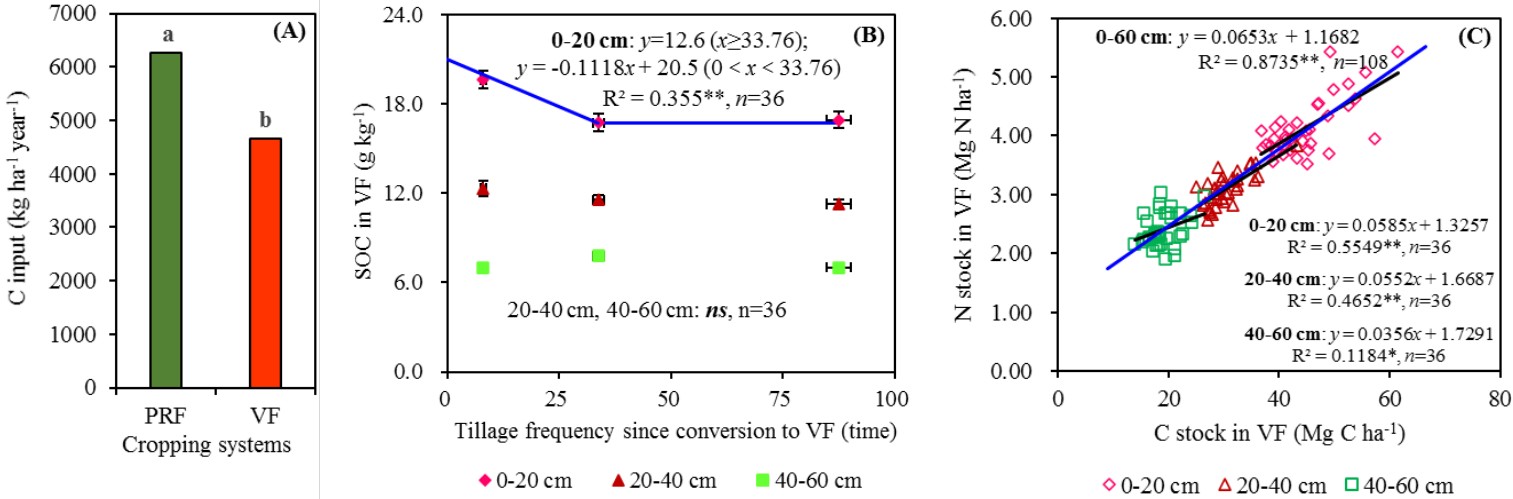

**Figure 2: Organic C inputs in paddy rice-oilseed rape rotation fields (PRF) and vegetable fields (VF) (A), the relationship between SOC concentration and accumulative tillage frequency since the conversion to VF (B), and the relationship between soil N stock and soil C stock in the 0-20, 20-40, and 40-60 cm soil layers in VF (C). Error bars represent standard errors. Significance levels \*, $P < 0.05$; \*\*, $P < 0.01$; \*\*\*, $P < 0.001$; ns, non-significant difference.**





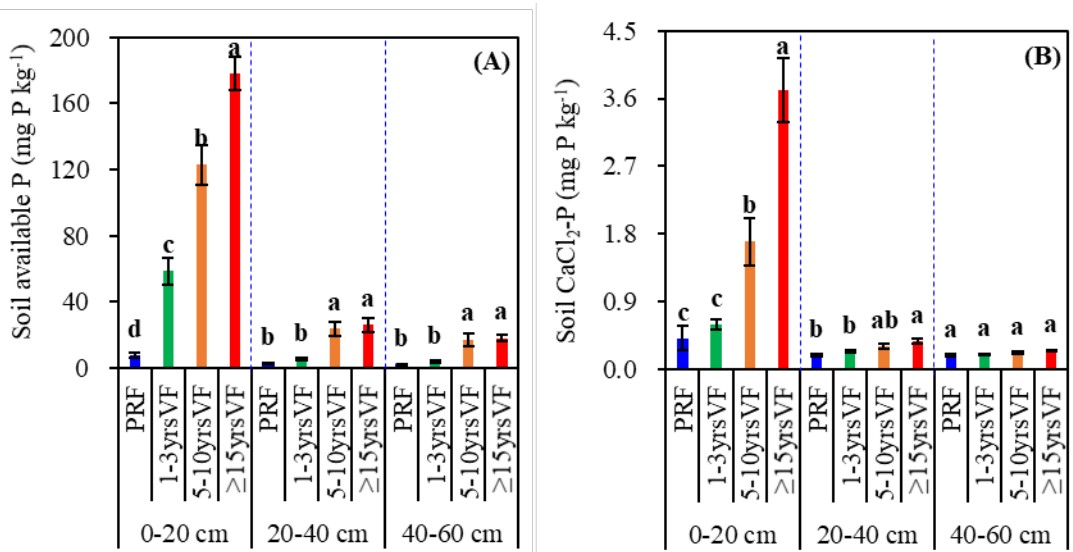

**Figure 3: Changes in Soil available phosphorus (P) (A) and CaCl₂-P (B) in the 0-20, 20-40, and 40-60 cm soil layers in paddy rice–oilseed rape rotation fields (PRF) and vegetable fields (VF) converted from PRF in southwest China 1-3, 5-10, or ≥ 15 years ago. For each soil layer, different lowercase letters indicate significant differences at *P*<0.05. Error bars represent standard errors.**



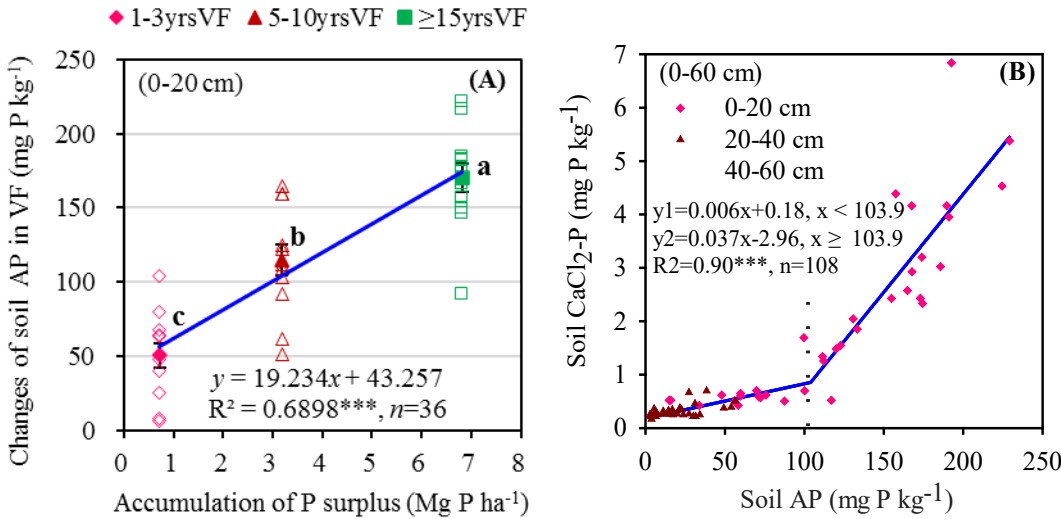

**Figure 4: The relationship between the change in soil available phosphorus (AP) before and after the conversion to vegetable fields (VF) and the accumulation of surplus P in the 0-20-cm soil layer (A), and the change point (dashed line) in the relationship between soil CaCl$_2$-P and AP in the 0-20 cm soil layer in VF (B). Different lowercase letters indicate significant differences at $P < 0.05$ among cropping systems (conversion to VF 1-3, 5-10, or $\geq$ 15 years ago). Error bars represent standard errors. ***, $P < 0.001$.**



**Figure 5: Soil water-soluble potassium (K) (A), water-soluble calcium (Ca) (B), water-soluble magnesium (Mg) (C), soil exchangeable (including the water-soluble fraction) K (D), exchangeable Ca (E), and exchangeable Mg (F) in the 0-20, 20-40, and 40-60 cm soil layers in paddy rice–oilseed rape rotation fields (PRF) and vegetable fields (VF) converted from PRF in southwest China 1-3, 5-10, or ≥ 15 years ago. For each soil layer, different lowercase letters indicate significant differences at $P<0.05$ among cropping systems. Error bars represent standard errors.**



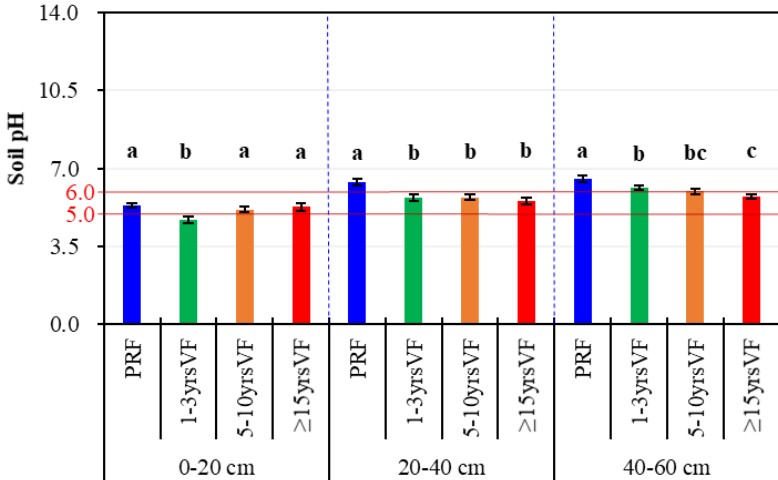

**Figure 6: Soil pH in the 0-20, 20-40, and 40-60 cm soil layers in paddy rice–oilseed rape rotation fields (PRF) and vegetable fields (VF) converted from PRF in southwest China 1-3, 5-10, or ≥ 15 years ago. For each soil layer, different lowercase letters indicate significant differences at $P<0.05$ among cropping systems. Error bars represent standard errors.**