# Peer review of "Significant soil degradation is associated with intensive vegetable cropping in subtropical area: A case study in southwest China"

_SOIL, 2021_

## Author Comment (AC1)

**Response to comments and suggestions by Editor/Reviewers**

**Reviewer #3: 'Comment on soil-2021-17', Anonymous Referee #3, 13 Apr 2021**

The present study intends to assess the effects over time of conversion from paddy rice-oilseed rape rotation (PRF) to vegetables (VF) cropping on soil organic carbon, total nitrogen, pH and nutrients considered as soil quality parameters. The subject fall within the general scope of the journal and it is a new and original contribution. In fact, this work provides useful data on soil organic matter within vegetables the cropping systems and confirms that is a sensitive indicator in environment degradation studies relative to land use and is worthy of publication in Soil. However, the manuscript requires major revision to improve the text and clarify some points before being acceptable for publication:

Response: Thanks for your kind comments and generally positive evaluation. We have revised the manuscript as per your suggestions. Please see below for details.

-Introduction section should include a work hypothesis.

Response: You are right, and we have added the corresponding hypothesis content to the Introduction section. "However, the impacts of land conversion from cereal cropping to vegetable production may be more adverse in tropical and subtropical regions compared to temperate regions due to higher temperatures and rainfall combined, low organic inputs and high frequency of tillage practices (NAPCC, 2016; Powlson et al., 2016; Sarker et al., 2018). Based on the background information, here we hypothesise that the conversion of cereals to vegetable production would results in significant degradation in agricultural soils in southwest China." Please see Line 49-53.

- In my opinion, the first objective is encompassed in the second objective because this previous survey is necessary to select the different VF farms according to the time elapsed since the conversion to vegetables as well as to discuss the differences found between both types of crops. In any case, this survey generates a database with little

data and regional or local relevance.

Response: Yes, we agree with the reviewer, and have revised the objectives accordingly. Please see Line 68-73, "The objectives of this study were to assess the impacts of land use conversion from paddy-rape rotation to vegetable production on soil pH, SOC, TN, soil C/N ratio, available P (AP), CaCl2-soluble P (CaCl2-P), and available base cations (K, Ca, and Mg). In the present study, paddy rice-oilseed rape rotation fields were used as a reference, because all vegetable fields in this region were converted from paddy soil. This will allow the comparison between the conditions on areas that changed to open-field vegetable production with surrounding areas that have remained under paddy rice-oilseed rape rotations."

- The experimental design is poorly explained, which combined with a poor presentation of statistical results, makes it very difficult to be certain what the researchers actually did. Were used plots randomly set up? For each time since conversion from PRF to VF; were all 12 plots located in the same farm? What is plot size? Where were 12 samples of PRF collected? Adjacent to what? Please add the distance between plots. Thus, information about plots distribution would help.

Response: We have revised Section 2.2 text carefully as per your suggestions: The field size of paddy-rape rotation and vegetable cropping were 0.03-0.13 and 0.01-0.08 ha, respectively." (please see Line 104-105). "First, we randomly collected 36 soil samples from a total of 133 commercial vegetable fields that were converted from paddy-rape rotation 1-3 years (n=12), 5-10 years (n=12), and ≥15 years (n=12) ago. Second, twelve paddy-rape rotation fields and the adjacent (< 100 m) vegetable fields were also randomly sampled for comparison. The soil type of the selected sites and the Local/regional climatic conditions and agricultural production practices were similar" (please see Line 108-112).

- Line 92: Why were the soil samples collected in September?

Response: To ensure the consistency of soil sampling, because both the paddy rice cropping and the first of three consecutive (non-heading) Chinese cabbage seasons are

generally harvested in September according to the Fig. 1B.

- The effects of type of crop and soil depth on measured variables could be tested by a two-way analysis of variance. This statistical analysis could support the discussion about the differences along soil depth detected between PRF and VF crops.

Response: We agree. As per your suggestion, we have added the two-way ANOVA results on the differences of soil properties along soil depth detected between PRF and VF crops. Please see Table 2 and Figure 3, 5 and 6.

- Please, provide information about methodology followed for analysing soil parameters.

Response: Accepted and done. Please see Section 2.3.

- Line 88: "To investigate the effects of long-term fertilization…." This is not correct because the soil samples were collected also from VF crops that were converted from PRF 1-3 years and 5-10 years.

Response: Thanks, we have revised the sentence as "To investigate the effect of conventional fertilization on nutrient surplus, enrichment/depletion, and leaching them into the soil profile of vegetable field, a multistage sampling technique was used to select representative fields for sampling soils from each cropping system." Please see Line 106-108.

- For the measurement of SOC following the dichromate digestion method soil samples are sieved to 2 mm and ground to a power-like consistency. However, the authors used soil samples passed through a 0.15 mm sieve for SOC and TN analyses, which prevents the comparison with other studies.

Response: Thanks for your meticulous mention. There many studies, which were mainly cited in the manuscript to make comparisons with the current study, have reported that soil samples could sieved to 0.15 to 0.25 mm (Yan et al., 2012; Wang et al., 2014; Gómez et al., 2020). Therefore, we think methodology and data generated in

this manuscript are comparable.

- Lines 154-156: Could you please add some more information in Mat & Meth. about tillage operations (e.g type of machinery, tillage depth, tillage timing)? a better description may help to support the suggestions made by the authors about the effects of tillage practices on soil organic carbon.

Response: Yes, thanks for your suggestions. We have carefully revised and added more details about tillage operations in M&M section. Please see Line 90-91.

- Line 117: the authors state that the accumulation of P surplus was calculated as the annual P surplus multiplicated by planting duration. I guess that they meant that annual P surplus was multiplicated by years since conversion to vegetable cultivation. This should be clarified.

Response: Yes, this has been clarified. Please see Line 144-146.

- Expressions like 1.92 or 0.80 mg P kg-1 (see abstract and through text) must be arranged by keeping the homogeneity of significant figures that your equipment generates. What is the precision of your method? Probably significant figures are "out" of the precision of your method!

Response: Here, we used a UV-Visible spectrophotometer method to determine the soil P (including AP and $CaCl_2$-P) concentrations, please see Line 124-127. This method is accurate enough to allow us to keep two decimal places.

- Some confusing sentences. What do the author's mean by "farmers´ survey methodology" or "local extension service"?

Response: We apologize for causing you any confusion, and have added some sentences to clarify this "farmers' survey methodology" and "local extension service". Please see Line 96-104 and Line 230-233.

- Lines 150-151: Where are the data of bulk density in the manuscript?

Response: We have added the bulk density data to the Table 2.

- Lines 212-221: This topic is not addressed in this study. Then, I suggest deleting it.

Response: Thanks for your suggestion. The discussion section in this paragraph is to further discuss the feasible strategies of improving the nutrient utilization efficiency and the soil quality of vegetable field in vegetable cropping systems. In our opinion, this is highly linked with the theme of this study.

- Lines 286-288: These two sentences are repetitive. One sentence should be deleted.

Response: Done. We have deleted the repeated sentence.

- The variance of data presented in Tables 1 and 2 should be expressed using standard deviation or standard error.

Response: Done.

- What data are shown in Table 1? Are average values of 36 surveys related to VF management?

Response: Table 1 is the investigated inputs and output for inventory analysis in paddy rice-oilseed rape rotation and vegetables production based on farmer surveys.

- Lines 165-166: The authors state that for every 100 kg ha-1 of P surplus in the VF, AP concentration in the 0-20 cm soil layer increased by 1.92 mg kg-1. However, according to the equation obtained by the authors for the data of the Figure 4 (y = 19.234 X + 43.257), for 100 kg ha-1 of P surplus in the VF, AP concentration in the 0-20 cm would increase by 45 mg kg-1.

Response: We apologize for this mistake, and have revised the sentence: "For every 100 kg ha$^{-1}$ of P surplus in the VF, AP concentration in the 0-20 cm soil layer increased 1.92 mg kg$^{-1}$". Please see Line 197-198.

- Lines 235-246: This topic is not addressed in this study. Then, I suggest deleting it.

Response: Accepted and corrected as requested.

\- Lines 255-258: In relation to inorganic N losses, the ammonium fixing capacity of the soils used (Ultisol) should be mentioned.

Response: We have added the details of the ammonium fixing capacity of the soils used (Ultisol) into the Section 4.2: "The slightly lower pH in subsoil in the vegetable field compared to that in the paddy-rape rotation might have slowed nitrification such that part of the inorganic N remained in the ammonium form sufficiently long enough for some immobilization to occur (Figure 6; Marschner, 2012). However, Nieder et al. (2011) indicated that the ammonium fixing capacity of the Ultisol, the type of soil used in this study, is extremely low. This may increase the risk of ammonium losses in this region, a potential environmental issue that warrants more attention in the future." Please see Line 285-290.

\- Line 266 (P enrichment): It has been omitted that when the soil pH reaches values less than 5.3 as occurred by conversion to VF, the presence of Al and Fe in clay minerals of Ultisols can contribute to fixing process of P.

Response: We agree and have added this sentence to the Line 304-306.

\- The English of the manuscript needs to be revised. They erroneously interchanged the term P surplus by surplus P throughout the text. For example, the authors write "the accumulation of surplus P" in the caption of Figure 4, whereas the title of x they write "accumulation of P surplus".

Response: We apologize for this unintentional omission and have carefully revised the scientific terminology as P surplus. Thanks.

Thank you very much for your consideration.

Kind regards,

(Ming Lu and Dunyi Liu)

---

## Author Comment (AC3)

**Response to comments and suggestions by Editor/Reviewers**

**Reviewer #2: 'Comment on soil-2021-17', Lukas Kohl, 30 Mar 2021**

Lu and co-authors report on a study that investigates the impact of transition that conventional rice paddy/oilseed rape to vegetable cultivation has on soil properties. They report that increased over-fertilization and decreased organic matter inputs lead to nutrient accumulation and organic matter depletion in subtropical agricultural soils. This is an important and timely study that fits well into the SOIL journal. The authors do not provide sufficient details on the methods to provide a full judgement on the validity of their results, but my overall impression is that their conclusions are well supported by the data. The manuscript is written in technically correct English, but I think that some sections need to be substantially improved before publication (see below). I therefore think that the authors should be given the chance to substantially improve the manuscript before publication (major revision or rejection with invitation to resubmit).

Response: Thank you very much for your kind comments and valuable suggestions for further improving the quality of current manuscript. We have revised the manuscript carefully, and all changes in the revised manuscript are made using Track Changes to make reviewing easy.

Specifically, I think that the quality of the different section of the manuscript varies a lot. In my opinion, the results and discussion sections are well developed and can be published with little modification, whereas introduction and methods sections need substantial improvement. The introduction is very short and does not provide a full overview about what is known about crop-to-vegetable transition impacts on soils. The text also misses 'flow' from one paragraph to the next. I would recommend to broaden the introduction, and look at crop-to-vegetable-transitions studies conducted throughout various climate systems. In addition, the introduction could explain how the different parameters measured are affected by this transition, and what results are

therefore expected. In the methods section, a lot more detail needs to be provided with regards to data sources and analytical methods.

Response: Many thanks, we accept all your comments and suggestions, and have revised and added more information to the manuscript, particularly in the Introduction and M&M sections, e.g. "On average, the sown area of non-grain crops has increased by >12% since 1980 in China, with a corresponding decrease in grain crop cultivation (Chen et al., 2016)" (please see Line 35-36), "Numerous studies have reported on the serious soil degradation in greenhouse vegetable fields which have been converted from maize-wheat rotation in the North China Plain, as a result of high total fertilizer inputs and increased frequency of soil cultivation, resulting in a significant decrease of soil C/N ratio and pH, enrichment of P and K, and salinization (Ju et al., 2007; Yan et al., 2012). However…" (please see Line 46-49), "However, there is no information available about the effects of land use changes from cereals to vegetable production on the concentration of SOC and TN in southwest China." (please see Line 58-59). In addition, in the M&M section, we have added a database of soil properties and provided further analytical parameters in the Supplementary Information. We now believe that the revised manuscript is much improved and is suitable for publication.

I have one major scientific concern that should be addressed by the authors. The manuscript states that soil properties of fields converted to vegetable farming were similar to those retained in conventional crop rotation. I think this authors need to provide more evidence for this point as it is an essential condition for the validity of their findings. In particular, I am wondering if the authors have any information whether the fields that were transitioned from conventional to vegetable farming were chosen randomly, of if farmers took soil properties into consideration when deciding if a given field was converted or not. If the latter were the case, the initial conditions of these field would be different from those retained in traditional crop rotation, which would limit the authors ability to link difference in soil properties to farming practices. This also applies to effects of time-since-conversion, since the farmer's considerations could change over time.

Response: The comparisons of paddy-rape fields with similar field following conversion to vegetable production are based on matched soil types, i.e. soils that have developed from the same parent material and under the same conditions of regional climatic conditions and agricultural production practices. We do not have any information on whether the fields that farmers transitioned from conventional to vegetable farming were chosen randomly. According to the farmer survey, the most important factor influencing the choice of land-use conversion is for economic benefit, because the output-input ratio of vegetables is significantly greater than that of cereal crops and oil crops. Additionally, factors such as limitation of agricultural irrigation conditions for paddy production, topography, and farm size may have influenced farmer choice of which fields to target for transition. Whilst we understand this reviewer's comment, the biggest differences between vegetable production and paddy-rape rotation are the nutrient inputs (including organic and inorganic, please see Table 1) and frequency of cultivation. So, we believe these factors would far outweigh any small variations in e.g. soil texture.

**Minor comments:**

L37: per unit nutrient inputs: Specify which unit you refer to (area?)

Response: Yes, it should be "per unit area of nutrient inputs". We have revised text in the new version of the manuscript.

L64-70: the data presented here should be supported by references.

Response: Thanks. We have added an appropriate reference in Line 80.

L81: farmers' survey methodology: I assume this refers to an established survey or similar undertaking, but this is likely lost in the translation, and comes across quite confusingly. I recommend clarifying this.

Response: Thanks, we have added a sentence to clarify this. "A survey of management practices was conducted by face-to-face interviews with farmers in Jinping County in

Guizhou Province. Based on farmers' survey methodology (Jia et al., 2013), two of the most important vegetable production townships in Jinping County, Xinhua and Dunzhai, were selected in this area. In each township, four villages were randomly selected, and 15-20 farmers from each village who had managed paddy-rape rotation before switching to vegetable cropping system were randomly surveyed. Survey questions were related to crop varieties and the management of the two cropping systems, including sowing rate, fertilizer (chemical or organic) application rate, tillage frequency, crop-residue management, crop growth cycle, cropping duration, and yield." Please see Line 97-103.

Section 2.3: A lot more detail is needed here. Ideally, a reference should be provided for each method, along with enough details such that the reader can reproduce the measurements.
Response: Accepted and done. Please see Line 118-129.

Section 2.4: Here as well more detail needs to be provided. What were the nutrient concentration in each fertilizer, and how were they obtained (source?).
Response: Accepted and done. We have added more details information in the Section 2.4, and have shown as listed in Table S1 and Table S2.

Section 2.5: The use of a two-segment fitted line in Fig 2b is inappropriate because there are only 3 independent datapoints (underlying replicates are not independent of each other). Essentially, this just connects two points with a line in between. I also don't think this contributes to the findings of the paper, so I would recommend removing it. Also, provide an overview on the amount if each fertilizer used in each treatment group.
Response: Thanks for your thoughtful analysis and suggestion. Actually, according to nonlinear regression analysis, the linear-with-plateau model produced the best fit for the plots of SOC concentration of 0-20 cm soil layer in the vegetable fields (VF) and number of tillage operations since conversion to VF based on 48 groups datapoints, which were hidden in the manuscript instead of these 3 independent datapoints (the

mean values for the corresponding grouping parameters). In addition, we would prefer to keep the corresponding Fig. 2b, as considering tillage frequency is the key factor affecting the decomposition of SOC in agricultural soils (Pires et al., 2016; Six et al., 2000), and quantifying the correlation between the change of SOC and tillage frequency is very important.

L152: I assume the unit here should be Mg ha-1 instead of Mg ha-1 yr-1
Response: Done.

L196: explain 'extension service'
Response: Accepted and corrected as requested. "The annual application rates of fertilizer N, P and K in the vegetable production (Table 1) were much higher than those recommended (N, 600-765, P, 79-144, K, 398- 498 kg ha$^{-1}$) based on crop requirements and soil properties, recommended by more than 30 agricultural research institutions nationwide (i.e. the extension service, Zhang et al., 2009)." Please see Line 230-233.

L207-8 weighted mean: weighted by what?
Response: We have clarified this sentence in Line 243-244.

286-288 check repeated statement.
Response: Done. We have deleted the repeated sentence.

Table 1 and 2: standard deviation or some other measure variance should state along with each value
Response: Accepted and done. We have added the standard deviation in the Table 1 and Table 2, respectively.

Fig 2. See comment on segmented fit above. Also, I guess with tillage frequency you mean number of tillage's since conversion? It would be simpler to just state number or years since conversion here. You could also add the PRF datapoint to the plot (time

since conversion = 0)

Response: Yes, you are right. We have corrected the statement of abscissa for Fig. 2B, and added the paddy-rape rotation datapoint to the plot.

Fig 3: Wider bars would make the figure easier to read.

Response: Done.

Fig 4: Did you actually calculate the cumulative P surplus for each site, or did you just multiply the average surplus with the number of years? If the latter, it would be better to just state years since conversion.

Response: Yes, we calculated the cumulative P surplus for each site.

Data availability: Raw data should be placed in a publicly available repository to meet the Copernicus/EGU data policy.

Response: Done.

Thank you very much for your consideration.

Kind regards,

(Ming Lu and Dunyi Liu)

---

## Author Comment (AC4)

**Response to comments and suggestions by Editor/Reviewers**

**Reviewer #1: 'Comment on soil-2021-17', Paul Hallett, 30 Mar 2021**

This is an extremely valuable paper. The authors provide a strong argument about socioeconomic drivers that have resulted in large increases in vegetable production. The environmental impacts are enormous, and it is clear that best farming practices are not being followed. A particularly useful aspect of this study is the use of commercial farms so that information is gleaned on both practice and impacts. The spatial coverage is impressive, providing a valuable dataset for a large body of potential follow-on research.

A general comment, which may complicate the study, is that the variability between farms is not described or discussed. The Tables could benefit from including either the range or statistical error in fertilizer application rates and soil properties. Are some farmers using much less fertilizer but achieving similar yields? Are some farmers using fertilizer application rates that are much higher than the averages presented, and what are the impacts?

Response: Thank you for your review and comments. We agree with you and have added the standard deviation in the Table 1 and Table 2, respectively. Additionally, because of the productivity differences caused by farmers' agronomic practices and soil properties, both of the scenarios you point out exist. Please see the following Figure. Although some farmers use much less fertilizer but achieve similar or higher yields, on average, the low regional vegetable yields and high levels of fertilizer input remain common. Our manuscript mainly describes the impacts from the change response of soil properties in vegetable fields. Currently, open-field vegetable-cropping practices in southwest China result in significant soil degradation and potential environmental pollution after conversion from a paddy rice-oilseed rape rotation system. Wang et al (2018b), who investigated the status quo of pepper production in the same subtropical region of China, indicated that not only nutrient management (in particular, decreasing application of N, P fertilizer), but also crop management (mainly planting density)

affected pepper yield. We have added these further discussion details to the Discussion section, please see Line 247-252.

Again, thank you very much for your attention and valuable comments.

[Figure]

**Figure** Relationships between nitrogen fertilizer production efficiency (**PFP-N, calculated as yield/N application rate**) and pepper yield (in fresh weight, **(a)**), non-heading Chinese cabbage yield (in fresh weight, **(b)**) in southwest China, respectively. The blue dotted lines represent the average production yield and PFP-N of all farmers surveyed. To analyze the effect of farm management on the environmental effects of pepper and non-heading Chinese cabbage production, the data on yield vs. PFP-N were divided into four groups: a low yield and low PFP-N group (LL); a low yield and high PFP-N group (LH); a high yield and low PFP-N group (HL); and a high yield and high PFP-N group (HH). Each hollow circle represents the data from one farm, and each solid circle represents the mean (with 95% confidence intervals) of the yield and PFP-N in each group.

There are some specific comments to address below, but this is generally a very well-prepared paper that is also extremely valuable.

Response: Many thanks. We appreciate these comments and suggestions from you, and we have revised the manuscript carefully, addressing all your comments and suggestions.

**Specific Comments:**

Unnecessary use of acronyms. Referring to the treatments as Vegetable and Paddy-Rape would be easier to follow.

Response: We agree, and we have revised the full text accordingly.

The survey, which is an important and novel component of this paper, is not mentioned.

Response: Thank you for this kind reminder. As the focus of this paper is the quantitative evaluation of changes in soil properties in vegetable fields, we did not include details of the farmer-based survey. However, as you and other reviewers' have suggested, we have provided additional detail of the survey methodology. Please see Line 17 and Line 97-103. Additionally, in term of the factor analysis in changes of soil carbon and pH, we have specifically analyzed the impact of farmers' agronomic practices. Please see Table 1 and 3, and Figure 2A and 2B.

Line 20 – is it due to decreased residue incorporation of lack of paddy cycle?

Response: Yes, it is one of the major reasons. Compared with the Paddy-Rape cropping system, the organic C input in the vegetable system decreased by 26% (Figure 2A). In addition, the higher frequency of tillage operations in vegetable production was another important impact factor because frequent tillage generally results in the break-up of soil macro-aggregates, soil structure damage, and an increase in soil aeration, which promotes microbial decomposition of SOC (Figure 2B).

The comparison between paddy and vegetable is not clear at the end of the Abstract.

Response: Thanks. We have revised part of the Abstract and emphasized on the comparison between paddy and vegetable systems.

Introduction

Line 37 – mixing up % and 'times' in the same sentence, which makes it harder to follow.

Response: Sorry for these errors, we have changed "%" to "times". Please see Line 40.

Line 40-44 – disconnected paragraph

Response: Accepted and addressed.

Line 48 – the impact of paddy production on C storage is not adequately described. There is a major change from this system to upland vegetable production.

Response: Thank you for your valuable suggestion. We have added more background information to carefully describe the innovation of this work: "For instance, because the paddy soil was flooded long-term and had two deep tillage operations per year with high inputs of carbon (Wang et al., 2014), while the vegetable fields generally had three to four deep tillage operations per year, resulting in greater disturbance of the surface soil layer (0-20 cm) in southern China. Wang et al. (2014) also found that the SOC concentration and the C/N ratio of soil in open-field vegetable systems converted from paddy fields decreased by 19.7% and 27.8%, respectively, which was mainly attributable to aggregate fragmentation." Please see Line 54-59.

Materials and Methods

For this study it is very important to characterize the soils, including their classification, dominant mineralogy and parent material. You should be able to get this from available soil survey data. You have described the soil as an Ultisol, but this is quite general. Readers need to be aware of the capacity of the soils to adsorb nutrients. Shallow soils affect some of this province, but I am not sure about this specific region. An idea of soil depth would help.

Response: We agree. We have revised the information about the regional soil, and added the soil depth details as per your suggestion, like "The main type of soil in this region is a typical Ultisol with loamy clay texture (average of sand 11.5%, silt 43.5%, and clay 45%, respectively) and alluvial parent material based on USDA soil classification system. Soil depth is generally >90 cm." Please see Line 82-84.

Line 89 – be clear that this is on commercial farms.

Response: Accepted and revised. Please see Line 108-109.

The experiment design and approaches are all good. You need to give details on how

bulk density was measured and be clear whether your 20 cm intervals incorporate the whole depth (which I assume it does) or just an interval defined by a core size.

Response: Thank you for your kind comment and suggestion. The bulk density was measured using the cutting ring method. Each single soil sample in this study incorporates the whole 20 cm depth. Please see Line 114-116.

Results

Although the data are presented clearly, this section could be more compelling. The amount of N application under vegetable production is staggeringly high. You could start by just mentioning N and stating the kg ha$^{-1}$ y$^{-1}$ amounts for different systems first and then use this to introduce high fertilizer use for other nutrients too. Table 1 gives no indication of variability, which is important to understand the commercial practices in place. If some farmers have much lower inputs, this is important to get across.

Response: Yes, annual total input of N in the vegetable cropping was 2.38 times higher than that of paddy-rape rotation, while the inputs of P, Ca, and Mg in the vegetable cropping were also many times greater than the paddy-rape rotation (i.e. 2.97, 4.40 and 7.14 times, respectively). Because all of these inputs are several times greater in the vegetable cropping systems, we have grouped them together and see no merit in separating these out.

Line 160'ish – the downward movement of P & K gives stronger reason to describe these soils more.

Response: Yes, we totally agree, and have provided additional soil descriptions as described in previous responses.

Discussion

This is excellent. I really like the start that describes what the farmers, which is then followed by the impacts.

Response: Thank you for your comments.

Line 223 – nice start! This is a good guide on what to do elsewhere to make the paper more compelling.

Response: Thank you so much!

Line 224 – whenever dealing with a soil depth, don't use 'higher' or it can get confusing.

Response: Ok, we understand and we have modified the text accordingly

Line 226 –I think not having a paddy cycle may be a big factor that is not being adequately considered.

Response: We agree with you. However, there was no paddy cycle in the annual vegetable production when converting from the paddy-rape rotation. Thus, the two factors resulting in further depletion of SOC in subsequent years following the conversion from paddy-rape to vegetable production, were low input of organic C from organic fertilizers and crop residues (Figure 2A), and a high tillage frequency (Figure 2B). We also discussed several possible management practices which should be conducted in the future to slow or reverse soil degradation, such as increasing organic inputs, optimizing fertilizer application, decreasing tillage frequency etc. Please see line 354-356 for details.

Line 242 – you are deviating away from vegetable production, where conservation agriculture may be less feasible. Only use practical solutions for the farming system. If CA is ok for vegetable production, you need to cite evidence for this rather than maize. What about better use of residue management from the vegetable crop or other practices? The discussion on alternative management strategies is weak. Can you obtain any further analysis from your data? Are there some farms using much lower fertilizer inputs who are maintaining yield or do they all apply very high rates of fertilizer? Can you do a simple cost-benefit based on the rising cost of N fertilizer vs. yield benefits?

Response: Thank you for your thoughtful analysis and valuable suggestion. Following the advice of Reviewer #3, we have removed this paragraph to retain focus avoid

confusion.

Line 261 – be clear if these studies were for the same region and/or soil type.

Response: Done. These studies were for the same production region. Please see Line 291-292.

Line 274 – Good argument on P impacts to rhizosphere, but do you think in a highly tilled, high nutrient system, AM will feature? You need to include more on the adsorption capacity of the soil in the region.

Response: Thanks. We do not think AM will feature in a highly tilled, high nutrient system. A high level of P enrichment in the soil is detrimental to plant growth because it inhibits the rhizosphere manipulation processes employed by plants to efficiently acquire P, including the colonization of roots by arbuscular mycorrhiza fungi and the exudation of organic acids or phosphatase enzymes. Please see Line 307-309.

Line 287 – remind readers of the pH value.

Response: Accepted and done.

Thank you very much for your consideration.

Kind regards,

(Ming Lu and Dunyi Liu)

---

## Author Comment (AC5)

**Cover Letter**

Jun. 1st, 2021

Dear Prof. García-Orenes,

Thank you for processing our manuscript entitled "**Significant soil degradation is associated with intensive vegetable cropping in subtropical area: A case study in southwest China**" (**soil-2021-17**) quickly. We appreciate the comments/suggestions from you and the reviewer on our paper.

We have revised the manuscript carefully, addressing all your and reviewer's comments and suggestions. Details of revision are given below. We also made a few minor edits in the manuscript to improve the language.

All changes made in the revised manuscript are done in Track Changes to make reviewing easy. We believe that the revised manuscript is much improved and is suitable for publication.

Thank you again and look forward to hearing from you soon.

Yours sincerely,

Dunyi Liu

College of Resources and Environment, Southwest University

Tiansheng Road, Chongqing 400716, China

Tel: 023-68251082; E-mail: liudy1989@swu.edu.cn